# Recent Advances in Studying In Vitro Drug Permeation Across Mucosal Membranes

**DOI:** 10.3390/pharmaceutics17020256

**Published:** 2025-02-14

**Authors:** Juan Song, Zizhao Xu, Lingxiao Xie, Jie Shen

**Affiliations:** 1Department of Biomedical and Pharmaceutical Sciences, University of Rhode Island, Kingston, RI 02881, USA; juan_song@uri.edu; 2Department of Pharmaceutical Sciences, School of Pharmacy and Pharmaceutical Sciences, Northeastern University, Boston, MA 02115, USA; zi.xu@northeastern.edu

**Keywords:** mucosa, in vitro permeation, physiological environment, locally acting, biorelevant, in vitro characterization-based bioequivalence

## Abstract

Transmucosal drug products, such as aerosols, films, semisolids, suppositories, and tablets, have been developed for the treatment of various human diseases and conditions. Transmucosal drug absorption is highly influenced by the biological structures of the mucosa and the physiological environment specific to the administration route (e.g., nasal, rectal, and vaginal). Over the last few decades, in vitro permeation testing (IVPT) using animal tissues or in vitro cell cultures have been utilized as a cost-effective and efficient tool for evaluating drug release and permeation behavior, assisting in formulation development and quality control of transmucosal drug delivery systems. This review summarizes the key mucosal permeation barriers associated with representative transmucosal administration routes, as well as considerations for IVPT method development. It highlights various IVPT methods, including vertical diffusion cell, flow-through diffusion cell, Ussing chamber, and transwell systems. Additionally, future perspectives are discussed, such as the use of optical methods to study in vitro drug permeation and the development of in vitro–in vivo correlation (IVIVC) for transmucosal drug development. The potential of IVPT as part of in vitro bioequivalence assessment strategies for locally acting transmucosal drug products is also highlighted.

## 1. Introduction

Mucosal membranes are commonly targeted for therapeutic delivery, owing to their rich blood supply and lack of a keratinized stratum corneum, which enable enhanced drug permeability and more efficient absorption of therapeutics for both systemic and localized drug actions compared to the skin [1]. Non-oral mucosal delivery routes (e.g., nasal, rectal, vaginal) offer significant advantages, such as bypassing first-pass metabolism and enzymatic degradation associated with oral drug delivery, enabling targeted delivery to the site of action, enhancing bioavailability, and reducing systemic side effects. These mucosal administration routes are particularly valuable for delivering proteins, hormones, and other therapeutics with poor oral bioavailability, as well as for diseases and conditions (e.g., bacterial vaginosis, ulcerative colitis) at the local sites of action [2,3].

The efficiency of transmucosal drug delivery is generally influenced by: (1) the biological structure and mucus composition of the mucosa, which vary across administration routes [4]; (2) the physicochemical properties of the drug, which determine its primary transport pathway (e.g., paracellular vs. transcellular) through the mucosal tissues; and (3) dosage forms or delivery systems utilized, which affect drug release and permeation behavior. Currently, transmucosal drug products approved by the U.S. Food and Drug Administration (FDA) are available in a variety of dosage forms, such as aerosols, films, semisolids, sprays, and tablets. Since these are considered complex drug products, it is critical to understand the impact of critical material and formulation parameters on drug permeation across the mucosa to facilitate the development of both innovator and generic transmucosal drug products.

Over the last few decades, in vitro permeation testing (IVPT) has been widely employed to assist in the selection of new chemical entities, formulation development, and to ensure consistent product performance of transmucosal drug products. IVPT typically involves the utilization of human or animal tissues mounted in diffusion cells, offering a cost-effective and efficient means of evaluating drug permeation behavior across biological membranes. This approach is instrumental in supporting formulation development, bioequivalence assessments, and regulatory review and approvals, particularly for locally acting semisolid drugs. Recent advances in IVPT methodologies also include the use of cell models and 3D-bioprinted tissues.

This review highlights recent progress in IVPT methodologies for evaluating drug permeation across mucosal membranes, particularly for locally acting transmucosal drug products. It provides an overview of the physiological environment and main transmucosal permeation barriers across different transmucosal routes and highlights various in vitro permeation techniques. The review also identifies key challenges and future perspectives for enhancing IVPT approaches to support drug product development.

## 2. Main Permeation Barriers of Representative Mucosal Membranes

Understanding the physiological environment and permeation barriers of mucosal membranes is essential for developing biorelevant in vitro methods to study drug permeation. Figure 1 provides an overview of the main permeation barriers associated with different mucosal membranes, and Table 1 summarizes their physiological characteristics.

### 2.1. Ocular Mucosa

Topical ocular drug delivery is a well-established approach for treating eye diseases affecting the eye surface and anterior chamber, such as inflammation and glaucoma. Drug absorption following topical ocular administration involves a complex process, including dilution in tear fluid (7–30 μL), diffusion across the mucin layer, penetration through the multilayered cornea, and transfer into the aqueous humor to reach the target sites such as the iris-ciliary body. While some drug absorption may occur through the conjunctiva/sclera, corneal absorption remains the primary route, particularly for diseases affecting the eye surface and anterior chamber [7]. However, short drug retention in the precorneal area due to the nasolacrimal drainage effect and the limited capacity (7–10 μL) of the cul-de-sac [17,18,19] may restrict ocular drug permeation and bioavailability.

The cornea, composed of epithelium, stroma, and endothelium, presents distinct drug permeation barriers: (1) Epithelium is lipophilic and composed of non-keratinized stratified squamous cells. Tight junctions, arranged in a “ribbon-like” structure, restrict paracellular diffusion, allowing better permeability for hydrophobic drugs while limiting hydrophilic drug transport [20,21,22,23]; (2) Stroma composed of a lamellar arrangement of collagen fibrils, accounts for 90% of corneal thickness and is highly hydrated, posing a significant barrier to lipophilic drugs but facilitating hydrophilic drug diffusion; and (3) Endothelium consists of a monolayer of leaky junctions, allowing the passage of macromolecules between the aqueous humor and stroma, which maintaining corneal clarity [24].

The mucin layer on corneal and conjunctival epithelia plays a protective role by restricting the penetration of foreign molecules and maintaining mucosal barrier integrity [24]. For instance, the presence of membrane-bound mucins has been shown to prevent corneal penetration of rose Bengal, an anionic organic dye used clinically to evaluate epithelial damage in conditions such as dry eye syndrome [5]. Although the gel-like mucin layer permits the diffusion of small molecules, its mesh size may restrict the transport of macromolecules and microbes. To prolong precorneal retention and enhance corneal penetration, mucoadhesive formulations are often designed to interact with the mucus layer, improving drug absorption and therapeutic effect [7].

A comprehensive understanding of the barrier properties and drug transport pathways within the cornea is critical for designing biorelevant IVPT methods to evaluate drug permeation. Effective models should incorporate tear fluid dynamics, mucin interactions, and layer-specific permeability to accurately predict in vivo performance and support formulation optimization of ocular drug delivery systems.

### 2.2. Nasal Mucosa

The nasal mucosa is a promising target for drug delivery due to its high vascularization, large surface area, and non-invasive accessibility. Intranasal drug products are commonly used to treat localized conditions such as nasal allergies, sinusitis, and congestion, providing rapid symptom relief with minimal systemic adverse effects [8]. Additionally, the olfactory region enables drug delivery into the central nervous system (CNS), making it an attractive pathway for neurological therapies [25,26].

The nasal respiratory mucosa consists of epithelium, basement membrane, and lamina propria. The epithelium is primarily composed of pseudostratified columnar epithelial cells, goblet cells, and mucous and serous glands, interconnected by tight junctions that restrict paracellular diffusion of hydrophilic drugs [27,28]. The pseudostratified columnar epithelial cells are further categorized into mucus secreting, ciliated, non-ciliated and basal cells. Ciliated and non-ciliated cells, covered with non-motile microvilli, provide a large surface area for drug absorption, while basal cells serve as progenitors within the collagen-based basement membrane, which overlays the lamina propria [26]. The nasal mucus layer, rich in mucin, may trap large molecular weight (MW) drugs (e.g., peptides and proteins) [29]. Moreover, mucociliary clearance affects drug residence and absorption efficiency [30]. Approximately 20–40 mL of mucus is secreted daily, covering extensive nasal mucosa surface area (160 cm^2^). This continuous mucus turnover can reduce drug residence time and potentially bioavailability [31]. Interestingly, the olfactory region in humans and primates contains minimal ciliated epithelium, leading to reduced drug clearance and prolonged drug retention [32,33], which is advantages for CNS drug delivery.

In order to develop biorelevant IVPT models for nasal drug delivery, it is essential to accurately mimic the nasal environment. Key factors to consider include mucin-drug interactions, mucus flow dynamics, mucociliary clearance mechanisms, and epithelial barriers along with drug transport pathways.

### 2.3. Rectal Mucosa

The rectum, the terminal portion of the large intestine, offers an alternative administration route for drug delivery with both local and systemic applications. Structurally, a normal adult human rectum is about 12–15 cm long with a surface area of 200–400 cm^2^ [34,35]. Unlike the small intestine, it lacks villi and microvilli, resulting in a limited absorptive surface area. The rectal epithelium consists of columnar cells and goblet cells that are responsible for secreting mucus, forming a 100 μm-thick layer that acts as a protective barrier and may hinder drug permeation [12,36].

Drug absorption across the rectal epithelium occurs through transcellular and paracellular routes [13,37,38]. However, several physiological factors affect drug permeation: (1) Low fluid volume (1–3 mL) limits drug dissolution, impacting drug absorption [36,39,40]; (2) Neutral pH with low buffering capacity, which may be altered by administrated drugs, potentially causing irritant; and (3) Presence of stool increases the viscosity of the rectal fluid, which may limit drug dissolution and mobility in the viscous rectal fluid [37,39,41]. Other factors such as rectal wall motility and formulation retention ability in the rectum also influence drug permeation and absorption [37,42]. For effective drug permeation and absorption through the rectal route, a drug or conjugate must strike a balance between lipophilicity (to cross the epithelial layer) and hydrophilicity (to ensure dissolution in the rectal fluid) [28]. Most rectal drug candidates are small molecules with MW < 500 g/mol and logP values ranging from 1.20 to 4.88.

To develop biorelevant IVPT models to assist in rectal drug development and quality control, it is essential to replicate the mucosal environment. Incorporating mucosal viscosity, and pH variability in IVPT systems may potentially improve predictive accuracy for in vivo performance, supporting formulation optimization and bioequivalence assessments.

### 2.4. Vaginal Mucosa

The vagina is a fibromuscular tube extending from the cervix to the vulva, measuring approximately 7–10 cm in length and 4 cm in width [43,44,45]. It is composed of four distinct layers: (1) Non-keratinizing stratified squamous epithelium; (2) Lamina propria (or tunica), which is rich in collagen, elastin, vascular and lymphatic channels; (3) Muscle layer with smooth muscle fibers arranged in both circular and longitudinal directions; and (4) Areolar connective tissue interwoven with a vascular plexus [34].

The vaginal epithelium consists of basal and parabasal layers, held together by tight junctions, desmosomes, and intermediate junctions [46]. Drug absorption involves two key steps: dissolution in the vaginal fluid followed by permeation through the epithelium. Previous studies indicated that most drugs permeated through the vagina via passive diffusion [47]. Vaginal fluid (~0.51 mL, pH 3.5 to 4.5), is composed of cervical mucus, glandular secretions, transudates, and endometrial fluids [16,48,49]. This fluid forms a thin mucus layer that influences drug absorption, bioadhesion and permeation, which could reduce the overall drug efficacy and residence time at the target tissue [48,50]. The mucus barrier provides greater resistance to large or charged molecules, while small, uncharged molecules diffuse more readily [51].

In order to assess the released drug mobility in the vaginal fluid and permeation mechanisms across the vaginal membrane, a biorelevant IVPT model should closely mimic the vaginal environment. Key considerations include the use of simulated vaginal fluid (to replicate the pH and mucosal composition) and biological membrane such as animal vaginal tissues, excised human vaginal tissues, or artificial vaginal membranes, to simulate permeation barriers.

## 3. Drug Permeation Mechanisms Across Mucosa

Mucoadhesion, drug dissolution, and epithelial penetration are critical steps for effective drug permeation and absorption through mucosal tissues. Drug dissolution and diffusion through the mucosal layer largely depend on the physiological environment (e.g., pH, volume, viscosity) of the mucosal tissues, the physicochemical properties (e.g., partition coefficient, MW) of the drug, and the formulation matrix and release kinetics. Mucus, composed of glycoprotein (mucin, MW: ~1–40 × 10^6^ kDa), lipids, salts, proteins, and carbohydrates, forms a hydrogel-like barrier that protects epithelial cells [4,52]. Mucin, primarily secreted by goblet cells, creates a mesh-like network that traps large molecules and restricts drug transport [53,54]. Effective permeation requires the ability of a drug to diffuse through mucus, equilibrate laterally, and partition into underlying tissues, typically driven by a concentration gradient [55]. Strategies to improve mucus penetration include the use of mucus-penetrating particles [56] and proteolytic enzymes to degrade mucin [57].

Drug penetration through epithelial tissues occurs via two main pathways: (1) Paracellular pathway which relies on passive diffusion through tight junctions and is primarily used by hydrophilic compounds (e.g., peptides). Molecules with radii greater than 15 Å have significant permeation limitations due to tight-junction selectivity. (2) Transcellular pathway is preferred by lipophilic drugs to diffuse across epithelial cell membranes [4]. Transcellular transport mechanisms include passive diffusion, carrier-mediated transport (active or facilitated diffusion) and endocytosis. Drug molecules move through passive diffusion along a concentration or electrochemical gradient without requiring energy. Unlike carrier-mediated transport, this process is not saturable and exhibits low structural selectivity [58]. Overall, drug lipophilicity, MW, ionization state, and dissolution rate strongly influence drug transport pathways and subsequent absorption. For instance, lipophilic molecules with MW less than 1 kDa are absorbed rapidly and efficiently via the transcellular pathway across the nasal mucosa [8]. Similarly, lipid solubility (e.g., carbonic anhydrase inhibitor) correlates directly with ocular permeation rates [59]. It has been reported that the acidic environment (pH 3.5–4.5) facilitates drug dissolution, however, the vaginal mucus may restrict permeation of ionized compounds [60].

## 4. IVPT Considerations and Systems

IVPT is a critical tool for assessing drug permeation using excised tissues or cell membranes applied in diffusion cells or Transwell systems. Since its first introduction in the mid-1970s, IVPT systems have undergone significant advancements [61]. Commonly used IVPT systems include vertical diffusion cell (VDC) apparatus, flow-through cell (FTC) apparatus, and Ussing chamber systems. Either an infinite or finite dosing strategy is implemented depending upon the study objective, with permeation rates estimated once a steady state is reached. In recent years, Transwell systems using cell layers and 3D tissue models have been developed to evaluate in vitro permeation behavior in vitro under biorelevant conditions. The receptor media used for IVPT of representative mucosae were summarized in Table 2.

### 4.1. Tissue-Based Models

For tissue models, it is critical to validate tissue integrity and thickness before and after IVPT studies to ensure reliable data. The IVPT duration has been reported to be up to 24 h for cornea, vaginal, rectal mucosa and 8 h for nasal mucosa [78,88,89,90]. In addition, tissue surface temperature should be carefully monitored and controlled throughout testing. Maintaining sink conditions is important to prevent saturation effects, particularly for poorly water-soluble drugs. Tissue variability must also be addressed by using tissues from multiple donors and conducting tests with sufficient replicates to ensure statistical robustness. Table 3 summarizes the tissue models commonly used for transmucosal permeation studies.

#### 4.1.1. VDC Apparatus

The VDC apparatus, originally developed by Franz [61], is widely used to evaluate in vitro performance, including in vitro release test (IVRT) and IVPT for topical drug products. It consists of two compartments: a donor cell and a receptor cell, with the tissue membrane mounted between them as the permeation barrier (Figure 2A). A water jacket is used to maintain a constant test temperature (e.g., 37 °C). A simulated body fluid or buffered solution is added to the receptor cell [47,83], while a test formulation is applied to the donor cell. Homogeneous mixing and proper hydrodynamics in the receptor cell are obtained using a magnetic stirring bar. The receptor cell features a sampling port for withdrawing aliquots at predefined time intervals, enabling time-course analysis and medium replacement to maintain sink conditions. Key parameters to be considered for IVPT using a VDC apparatus include: dosing amount, effective permeation area, receptor volume, and hydrodynamic conditions in the receptor cell [109]. These parameters must be carefully selected and controlled to minimize IVPT data variability and ensure reproducible and reliable results. While the VDC system is simple to use, a potential limitation is the decreasing of diffusion gradient caused by the accumulation of the permeating drug in the receptor compartment [88]. Strategies to maintain sink conditions such as replacing and replenishing receptor media at each sampling time point, as well as optimizing sampling intervals can help mitigate this issue.

#### 4.1.2. FTC Apparatus

The FTC system features a vertical arrangement similar to the VDC apparatus (Figure 2B), but includes input and output ports in the receptor cell to enable continuous fluid flow. This design better simulates in vivo conditions compared to the static VDC apparatus, as it effectively maintains sink conditions and mimics blood flow beneath biological membranes [110]. Flow rate is a critical parameter that influences drug permeation rates in the FTC apparatus. Depending upon drug solubility, low flow rates may lead to drug saturation in the receptor fluid, thus violating sink conditions and reducing permeation rates for highly permeable drugs. Conversely, high flow rates may cause excessive dilution, potentially underestimating permeation rates for low-permeability drugs [111]. The FTC system offers numerous advantages, including auto-sampling, easy maintenance of sink conditions, and simulation of physiological conditions such as blood flow beneath biological membranes [110]. However, the pumped volume must significantly exceed the receptor’s volume to thoroughly wash the permeating substance. Therefore, the receptor chamber volume should be kept small to manage the effluent volume effectively [88].

#### 4.1.3. Ussing Chamber System

The Ussing chamber system, introduced by Hans Ussing in the early 1950s [112], allows precise control of temperature and oxygenation while simultaneously measuring electric parameters to monitor tissue viability during IVPT studies [113]. The Ussing chamber system is classified into two main types: circulating chambers and continuously perfused chambers. The Ussing chamber system typically consists of two main functional components: a chamber unit that houses the biological tissue and maintains hydrostatic pressure to prevent tissue damage caused by bending [114]; and an electrical circuit that measures current (I) and voltage (V) to calculate resistance (R) and more complex parameters such as impedance and capacitance [115]. The circulating chamber features a U-shaped tubing system filled with carbogen gas (95% of O_2_ and 5% of CO_2_) and N_2_ to maintain constant stirring and ensure mucosal viability [88]. Ussing chambers are available in vertical (Figure 2C) and horizontal configurations, tailored for specific experiments. Key parameters to be considered for the Ussing chamber system include temperature, gassing, electrical interferences (noise), and offset voltages [115]. While the Ussing chamber system can support both native human or animal tissue and cell models [113], the viscosity of semisolids or dissolved solutes may impair gaseous and nutrient exchange, affecting cell viability and data accuracy [88].

### 4.2. Cell-Based Models

Cell-based models utilizing immortalized human cells or primary cell cultures assess drug permeation behavior and biocompatibility [116]. These cell-based models can reduce the reliance on animal tissues and address translational challenges while enabling reproducible and ethically accepted methods in early drug development. Transepithelial electrical resistance (TEER) measurements are used to monitor the integrity of tight junctions and mucosal barriers [81]. However, simplified cell models may not fully replicate the complex tissue environment, which limits their capacity for formulation evaluation [117].

#### 4.2.1. Transwell System

The transwell system (Figure 2D), equipped with polycarbonate or polyester membrane inserts cultured with cells, has been used for drug permeation studies [38,81]. Cell monolayers, acting as permeation barriers, typically take several days to form after cell seeding, and TEER measurements ensure monolayer integrity. Inserts with intact cell monolayer layers are transferred to the receiving wells containing a receptor medium (e.g., simulated vaginal fluid) and a test formulation is added to the insert/donor chamber. Samples are collected from receiving wells at predetermined time points and replaced with fresh media. It is critical to monitor TEER during IVPT to ensure cell monolayer integrity throughout the study [81]. Table 4 provides an overview of cell models reported for transmucosal IVPT using transwells. While transwell models reduce animal use and support mechanistic studies, experimental variability can arise due to culture conditions and poorly characterized lipid compositions [118].

#### 4.2.2. Corneal Cell Models

Primary human corneal epithelium (HCE) cells are not ideal for large-scale permeation studies, since primary cells cease to grow after a few passages in culture. As a result, immortalized human corneal epithelium cell lines such as HCE-S and SV 40 HCE-T are typically preferred for such studies, as shown in Table 4 [121,134]. HCE-S cells [119,135] can be readily cultured in basic DMEM and serum-based media, offering a distinct advantage over existing corneal epithelial cell lines. Although SV 40 HCE-T has been widely applied for in vitro models of HCE cells, the genomic content of HCE-T cells appeared to be different from a normal healthy genome, and they also showed a significant number of heterogeneous cell populations based on an array-based comparative genomic hybridization analysis [135]. On the other hand, tetracycline-responsive HPV 16-E6/E7-transduced HCE cell clones [121] exhibited tightly controlled inducible proliferation, relatively normal differentiation of the corneal epithelium, and characteristic HCE morphology. More recently, a human organotypic cornea with typical corneal structures was developed, which contains a single layer of endothelial cells and a multilayered epithelium with flat superficial cells. However, the barrier functions of the human organotypic cornea still need to be thoroughly evaluated.

#### 4.2.3. Nasal Cell Models

Immortalized nasal cell lines have been shown to have high proliferation rates, greater reproducibility, and lower costs, and are easier to maintain in culture compared to primary cell cultures [123,136]. RPMI 2650 is the only cell line derived from human nasal tissue; however, it has several limitations: (1) it lacks tight-junction expression; (2) it does not differentiate into goblet or ciliated cells; and (3) it fails to reach confluence [137,138]. Although RPMI 2650 cells have been used in some permeation studies, results were highly dependent on culture conditions. It was reported that the air–liquid interface culture of RPMI2650 cells could be suitable for in vitro permeation studies across nasal mucosa [71].

#### 4.2.4. Rectal Cell Model

The Caco-2 monolayer model has been the main model for studying drug permeation through the rectal membrane. For example, rectal delivery of polymeric nanoparticles containing an anti-HIV drug, dapivirine, was evaluated using Caco-2 monolayers. The retention of the drug within the Caco-2 cell monolayers and pig mucosa showed a similar trend for different nanoparticle formulations, indicating in vivo relevance [81].

#### 4.2.5. Vaginal Cell Models

Ectocervical cells appear to be the most interesting in vitro model for studying vaginal drug permeability histologically. Human ectocervical epithelial (hECE) cells have been shown to maintain most of the phenotypic and biological features of the human cervicovaginal epithelium in vivo, including intercellular tight junctions and polarization of apical and basolateral surfaces. However, a significant lack of human ectocervical explants limits this model’s application. In contrast, several continuous human cervicovaginal cell lines, which are easier to maintain than primary cells, are more readily accessible. Cultured cervical cell lines such as CaSki, ECE16-1, and HT3 have been shown to have leaky epithelia based on both TEER and pyranine permeability studies, resembling the epithelia derived from human explants [126,139]. The commercially available EpiVaginal^®^ model composed of untransformed human vaginal ectocervical (VEC) epithelial cells and nonepithelial elements such as dendritic cells and fibroblasts, presenting unique organotypic characteristics, has been utilized for IVPT of various drugs [123]. However, this model lacks a mucosal component and is expensive, which may limit its applicability. HVE^®^ is composed of A431 cells derived from a vulval epidermoid carcinoma cultivated on an inert polycarbonate filter at the air–liquid interface in a chemically defined medium. This model has been shown to be histologically similar to the in vivo vaginal mucosa and is used to screen and assess active ingredients or formulations for vaginal administration [88].

It is critical to note that research involving human and animal tissues requires review and approval or exemption from an institutional review board (IRB). In the United States, IRB exemption is generally granted for research using unidentifiable human tissue samples from cadavers. However, this may differ across countries. Nevertheless, researchers must carefully review tissue sources and material transfer details with their IRB and complete the appropriate human research ethics training according to institutional and national requirements. Additionally, research involving cells must be reviewed and approval by an institutional biosafety committee (IBC) to ensure compliance with relevant biosafety guidelines and to protect public health and the environment. Adherence to both biosafety and human research ethics guidelines is essential for conducting responsible and ethical research.

### 4.3. Data Analysis

Transmucosal diffusion is generally a passive process driven by a concentration gradient. Initially, drug molecules must be released from the formulation vehicle to reach the mucosal interface, after which they partition into the upper layers of the mucosal tissue. A key parameter influencing the permeation rate is the drug’s lipid–water partition coefficient.

For tissue-based models, the concentration gradient across the mucosa may not be linear at the start of IVPT. Over time, however, a steady state is achieved, at which point Fick’s first law of diffusion can be applied to describe drug transport across the mucosal barrier. Fick’s first law relates to the amount of solute (dosed drug, *Q*), membrane area (*A*), time period (*T*) regarding the concentration gradient (Δ*C_s_*), the diffusion coefficient of the mucosal membrane (D), and the transport path length (*h*) (Equation (1)). A steady-state flux, *J_ss_*, can then be defined (Equation (2)). Higuchi suggested that flux is more appropriately expressed in terms of thermodynamic activity, rather than relying on a concentration approximation [140]. The constant maximum flux occurs when the solute (drug) solubility reaches the maximum value (*S_s_*) in the tissue (Equation (3)). The thermodynamic activity of the solute (dosed drug) is generally defined as the fractional solubility of the solute (drug) in the tissue (*C_s_*/*S_s_*), which can also be expressed in terms of the solute (drug) concentration in the vehicle (formulation, *C_v_*) and the partition coefficient *K* (*K* = *C_s_*/*S_s_*, Equation (4)) [141].

The in vitro permeation rate is typically calculated by plotting the cumulative amount of drug permeated per unit of surface area against the square root of time. The slope of the regression line (derived from the linear portion of the curve) represents the permeation rate [38].(1)$$Q=\frac{DAT\Delta C_{S}}{h}\;or\;Q=\frac{C_{n}\cdot V+(\sum_{1}^{n-1}C_{n-1})\cdot V_{r}}{A}$$(2)$$J_{ss}=\frac{{\Delta Q}_{t}}{\Delta t\cdot A}$$(3)$$J_{max}=\frac{DS_{S}}{h}$$(4)$$J_{ss}=\frac{KDC_{v}}{h}$$
where, *C_n_* is the drug concentration at the respective time point, *C_n_*_−1_ is the drug concentration at the previous time point, *V* is the medium volume in the receptor cell, *V_r_* is the sampling volume, and *A* is the permeation area. Δ*Q_t_* is the difference in the drug amount permeated between two time points (Δ*t*), and *C*_0_ is the initial drug concentration in the donor chamber.

Beyond the steady state, a transient model can be used to describe time-dependent drug permeation. Fick’s second law is used as a basic model for this, assuming that the tissue membrane is a pseudo-homogeneous membrane and that both drug partition coefficient and diffusion coefficient remain constant over time (Equation (5)). Various modeling approaches, including compartmental and complex models, have been developed to account for these dynamics [141,142].(5)$$\frac{\partial C}{\partial t}=D\frac{\partial^{2}C}{\partial x^{2}}$$
where, *C* is the concentration of the permeating solute (drug) at time t and depth *x* within the tissue membrane.

For cell-based models, the apparent permeability coefficient (*P_app_*) is calculated using the following equation:(6)$$P_{app}\;(cm\cdot s^{-1})=\frac{Q}{ACt}$$
where *Q* is the total amount of drug permeated into the receptor compartment (μg), *A* is the diffusion area (cm^2^), *C* is the initial drug concentration in the donor compartment (μg/cm^3^), and *t* is the total duration of IVPT [38].

## 5. Future Perspectives

IVPT has long been instrumental in studying transmucosal permeation mechanisms and supporting drug product development. With significant advances in biomedical sciences and biotechnology, innovative techniques have emerged to enhance our understanding of transmucosal drug permeation. Among these, non-destructive optical methods, particularly confocal laser scanning microscopy (CLSM) and confocal Raman spectroscopy and near-infrared (NIR) spectroscopy, have shown great promise in studying the permeation of topical drug products. These optical methods can provide more detailed information on functional groups of drug compounds. For instance, CLSM in combination with other techniques such as transmission electron microscopy and scanning electron microscopy has been utilized to explore the corneal penetration pathways of labeled peptides and to examine changes in cytoskeletal structure and tight-junction permeability when exposed to penetration-enhancing agents [143,144,145]. CLSM has also been employed to identify the transcellular routes through which various nanocapsules penetrate the cornea, both ex vivo and in vivo [146]. In terms of nasal drug permeation, CLSM has provided valuable insights into the permeation and uptake mechanisms of therapeutic proteins, such as human calcitonin and insulin, through ex vivo nasal epithelia [147,148]. CLSM has also been utilized in elucidating the impact of size and polyethylene glycol (PEG) coating density on the transport of poly(lactic acid)–poly(ethylene glycol) (PLA–PEG) nanoparticles and microparticles across the nasal mucosa in rats [149]. Oranat et.al used confocal Raman spectroscopy to measure the local concentrations of the topical vaginal anti-HIV microbicide (tenofovir) in fluids, drug delivery gels, and tissues. Their findings demonstrated a strong correlation between tenofovir tissue permeation obtained using confocal Raman spectroscopy and that obtained using LC-MS/MS [150]. Furthermore, the combination of confocal Raman spectroscopy and optical coherence tomography (OCT) has enabled temporally and spatially specific measurements of tenofovir concentration across the full depth of excised porcine rectal mucosa [151]. Despite its potential as a minimally invasive and label-free analytical tool for determining drug concentrations during the permeation process, confocal Raman spectroscopy has certain limitations: (1) high variability in tissue composition and structure, which can complicate Raman spectral analysis [152]; and (2) ongoing diffusion and changes in tissue hydration during IVPT, which can alter Raman spectra. As a result, confocal Raman spectroscopy is still considered a semi-quantitative technique.

As drug delivery systems become more complex, IVPT is expected to evolve to include more sophisticated models, such as organ-on-a-chip systems, to replicate human physiological conditions more effectively. Recent efforts have focused on building models, such as a nasal epithelial mucosa, to mimic nasal tissue physiology in vitro [153,154] for drug transport studies of aerosols [155]. Advanced tools such as the dynamic micro tissue engineering system (DynaMiTES) [156] were used to build a 3D hemi-cornea construct composed of human keratocytes (HCKeCa) and epithelial cells (HCE-T) [157]. Although this ocular tissue-on-a-chip model was developed to study the disease progression of macular degeneration, it could potentially be useful for studying drug permeability [158,159].

In vitro–in vivo correlation (IVIVC) is defined by the U.S. FDA as a predictive mathematical model describing the relationship between the in vitro properties of a dosage form and its relevant in vivo response. While IVIVC has been extensively developed for extended-release oral drug products, regulatory agencies currently do not permit the use of IVIVC as a surrogate for bioequivalence (BE) studies of locally acting transmucosal drug products. Recent advancements have demonstrated the feasibility of developing an IVIVC for dermatological products such as transdermal patches [160]. For most percutaneous drug products, the rate-limiting step in absorption is drug transport across the stratum corneum (SC), rather than its release from a transdermal formulation. Similarly, the epithelium of mucosae may serve as the primary barrier to drug absorption in the mucosal membrane. This suggests that IVPT studies using exercised tissues with good integrity could potentially correlate well with in vivo absorption of transmucosal drugs. For systemically acting drug products, BE, required for generic drug approval, is evaluated based on plasma pharmacokinetic approaches, which can provide accurate and direct evidence through area under the plasma concentration curve (AUC) and maximum concentration (*C_max_*). However, BE assessment for locally acting topical drug products often relies on costly comparative in vivo clinical endpoint studies, which can be challenging [161]. Recognizing these challenges, the U.S. FDA has made significant efforts in recent years to provide guidance documents supporting in vitro characterization-based BE approaches to demonstrate BE for topical dermatological drug products [162,163,164,165,166]. IVPT is a key component of the in vitro characterization-based BE approaches recommended in several product-specific guidance (PSG) for topical drug products to compare prospective generic products and reference drug products [167,168,169,170,171]. The endpoints of IVPT are based on parameters that characterize the rate (flux, *J*) and extent (total cumulative amount, AMT) of drug permeation. The maximum flux (*J_max_*) at the peak of the drug flux profile and the AMT should both be compared for locally acting test topical products and reference standards (RSs). This is somewhat analogous to the comparison of the *C_max_* and AUC for systemically acting test products and RSs [172]. Similar in vitro characterization-based approaches are currently under investigation for locally acting semisolid dosage forms administrated via mucosal routes, such as vaginal and rectal. Early evidence supports the feasibility of in vitro characterization-based BE approaches for locally acting transmucosal drug products [173,174,175].

In silico prediction of mucosal drug permeation may be an alternative to experimental permeation studies in the future. Various in silico models, such as the Potts–Guy model, have been developed to correlate transdermal permeability coefficients and drug molecule characteristics (such as lipophilicity and MW) [121]. Although there is currently no simple model to predict mucosal drug permeation, it is expected that this will be addressed in the near future with advances in physiologically based pharmacokinetic (PBPK) modeling.

## 6. Summary and Outlook

IVPT has emerged as a pivotal tool in evaluating the performance of transmucosal drug products, offering insights into their absorption mechanisms and predicting in vivo drug behavior. The integration of IVPT with IVIVC for locally acting drug products, including those administered through the mucosal routes, holds significant potential for advancing regulatory science and streamlining drug approval processes. However, challenges remain, including variability in mucosal tissue properties and the need for standardized IVPT protocols. Continued refinement of more biorelevant and robust IVPT methodologies, coupled with advanced analytical tools and deeper insights into drug permeation mechanisms, is expected to drive the development of safer and more effective transmucosal drug products. As these methods become more standardized and validated, they are likely to play a crucial role in the FDA’s regulatory strategies, facilitating the development and approval of both innovator and generic transmucosal drug products, which will ultimately improve public access to medications.

## Figures and Tables

**Figure 1 pharmaceutics-17-00256-f001:**
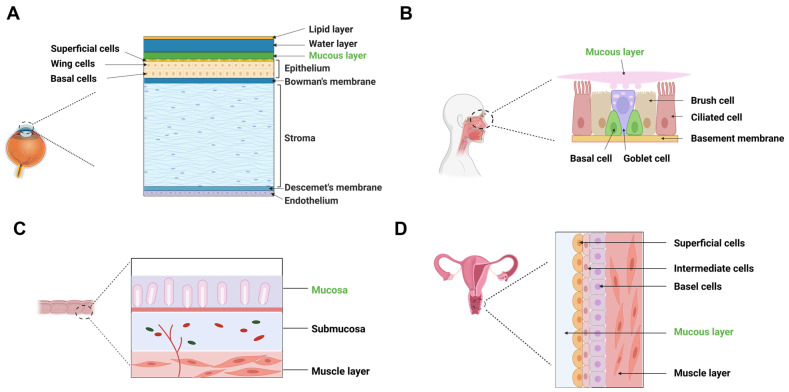
Main permeation barriers of representative mucosal administration routes. (**A**) cornea; (**B**) nasal mucosa (respiratory region); (**C**) rectal mucosa; (**D**) vaginal mucosa.

**Figure 2 pharmaceutics-17-00256-f002:**
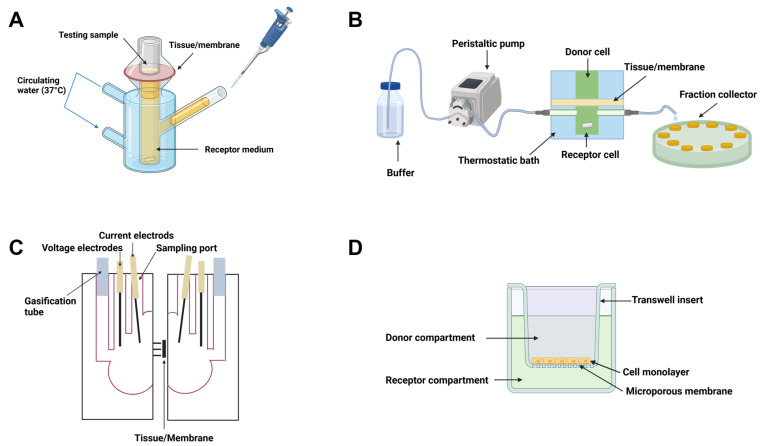
Schematic diagram of tissue-based (**A**) vertical diffusion cell (VDC), (**B**) flow-through vertical cell apparatus (FTC), (**C**) vertical Ussing chamber apparatus and (**D**) transwell system.

**Table 1 pharmaceutics-17-00256-t001:** Physiological environment of different mucosa.

Mucosa	Surface Area	Thickness	pH of the Fluid	Volume of the Fluid
Corneal	1.04 cm^2^ [5]	520 μm [6]	~7.4	7–30 μL [7]
Nasal	130 cm^2^ (respiratory) [8]	0.3–5 mm (respiratory) [9]	5–6.5	75–135 μL [10]
Rectal	200–400 cm^2^ [11]	100 µm [12]	7.2–7.4 [13]	1–3 mL [11]
Vaginal	360 cm^2^ [14]	150–200 μm [15]	3.5–4.5	0.51 mL [16]

**Table 2 pharmaceutics-17-00256-t002:** The receptor media used in transmucosal IVPT studies.

Mucosa	Simulated Fluids in Receptor Cell	Components of Simulated Fluids in Receptor Cell	pH
Corneal	Glutathione bicarbonate Ringer (GBR) buffer [62,63]	NaCl (110 mM), KCl (2 mM), K_2_HPO_4_ (1 mM), MgCl_2_ (0.6 mM), calcium gluconate (1.4 mM), glucose (5 mM), reduced glutathione (0.3 mM), sodium gluconate (15 mM), and NaHCO_3_ (28.5 mM), CO_2_ (5%)	6.85
	Balanced salt solution (BSS) buffer solution with 10 mM HEPES [64]	NaCl 0.64%, KCl 0.075%, CaCl_2_·2H_2_O 0.048%, MgCl_2_6·H_2_O 0.03%, C_2_H_3_NaO_2_·3H_2_O 0.39%, C_6_H_5_Na_3_O_7_·2H_2_O) 0.17%, NaOH and/or HCl to adjust pH	7.4
	PBS [65,66]	phosphate buffer saline	7.4
	PBS [67]	phosphate buffer saline	7.0
	EBS buffer solution [68]	Earl’s balanced salts (EBS)	7.4
	BSS buffer [69]	NaCl 0.64%, KCl 0.075%, CaCl_2_·2H_2_O 0.048%, MgCl_2_6·H_2_O 0.03%, C_2_H_3_NaO_2_·3H_2_O 0.39%, C_6_H_5_Na_3_O_7_·2H_2_O) 0.17%, NaOH and/or HCl to adjust pH	7.4
Nasal	Transport medium (TM) [70]	NaCl (136.89 mM), KCl (5.36 mM), Na_2_HPO_4_ (0.34 mM), KH_2_PO_4_ (0.44 mM), MgSO_4_·7H_2_O (0.41 mM), glucose (19.45 mM), CaCl_2_ (1.26 mM), MgCl_2_·6H_2_O (0.49 mM), NaHCO_3_ (4.17 mM), and HEPES (10.00 mM).	7.4
	Kreb’s bicarbonate Ringer’s solution (KBR) [71,72,73]	CaCl_2_ (1.25 mM), KCl (4.7 mM), KH_2_PO_4_ (0.6 mM), MgSO_4_·7H_2_O (1.2 mM), NaCl (108 mM), NaHCO_3_ (16 mM), Na_2_HPO_4_ (1.8 mM), d-glucose (11.5 mM), fumarate (5.4 mM), glutamate (4.9 mM) and pyruvate (4.9 mM)	7.4
	Simulated nasal electrolyte solution (SNES) [74]	7.45 mg/mL NaCl, 1.29 mg/mL KCl and 0.32 mg/mL CaCl_2_·2H_2_O	5.5
	PBS with 5% DMSO (*v*/*v*) [75]	phosphate buffer saline added with 5% DMSO (*v*/*v*)	7.4
Rectal	0.1× PBS [76]	phosphate buffer saline	7.2
	PBS [77,78,79]	phosphate buffer saline	7.4
	Phosphate buffer solution [80]	50 mL phosphate buffer solution	7.4
	HBSS with poloxamer 407 [81]	Hank’s balanced salt solution (HBSS), added with 0.2% (*w*/*v*) poloxamer 407	7.3
	Saline [82]	0.9% *w*/*v* NaCl	N/A
Vaginal	PBS [47,83,84,85]	phosphate buffer saline	7.4
	HBSS with poloxamer 407 [81]	Hank’s balanced salt solution (HBSS), added with 0.2% (*w*/*v*) poloxamer 407	7.3
	Ringer buffer [86]	7.5 mL, oxygenated Ringer buffer at pH 7.4 containing 10 mM glucose, 0.3 g/L L-glutamine, and 2% albumin (*v*/*v*)	7.4
	Simulated vaginal fluid [87]	NaCl (3.51 g/L), KOH (1.4 g/L), Ca(OH)_2_ (0.222 g/L), acetic acid (1 g/L), lactic acid (2 g/L), glycerol (0.16 g/L), urea (0.4 g/L), and glucose (5 g/L)	4.2

N/A: no information provided from the literature.

**Table 3 pharmaceutics-17-00256-t003:** Tissue models commonly used in transmucosal IVPT studies.

	Mucosa
Corneal	Goat cornea [91], rabbit cornea [66,92], porcine cornea [93]
Nasal	Human biopsies from posterior part of inferior turbinate of normal patients [94], bovine nasal mucosa [68,95,96,97], porcine respiratory mucosa [98,99], rodent nasal mucosa, ovine nasal mucosa [100,101,102], rabbit nasal mucosa [101,103,104]
Rectal	Cattle and rat rectum [80,105], porcine rectum [76]
Vaginal	Porcine vaginal tissue [83], bovine vaginal tissue [86], sheep vaginal tissue [106,107,108], human cervical and vaginal tissue [47]

**Table 4 pharmaceutics-17-00256-t004:** Representative cell models reported for transmucosal IVPT using a transwell.

Mucosa	Cell Model	Source
Ophthalmic	HCE-S [119,120]	Corneal epithelial (spontaneous)
	SV 40 HCE-T [120]	Simian virus 40 immortalized human corneal epithelial cells
	Human tet HPV16-E6/E7-transduced HCE [121]	Tetracycline-responsive human papillomavirus (HPV) 16-E6/E7-transduced human corneal epithelial cell clones
	Human organotypic cornea equivalent [122]	Immortalized epithelial (CEP-17-CL4) and endothelial (HENC) cells and primary human stromal cells
Nasal	RPMI 2650 [70]	Human nasal epithelial tissues
Rectal	Caco-2 cell lines (human colon carcinoma) [123,124]	Human colon carcinoma
Vaginal	hECE (primary) [125,126]	Human ectocervical epithelial
	HEC-1A (immortalized) [127]	Uterine endometrial adenocarcinoma
	CaSki (immortalized) [81]	Endocervical carcinoma cells
	C-33A (immortalized) [128]	Cervical cells
	ECEI6-1 (immortalized) [125,126]	Ectocervical carcinoma cells
	HT3 (immortalized) [125,126]	Cervical carcinoma cells
	ME-180 (immortalized) [129]	Endocervical cells
	SiHa (immortalized) [129]	Uterine cervix
	EpiVaginal^®^ (cell-based tissue model) [130]	Normal, human-derived vaginal ectocervical epithelial and dendritic cells
	HVE^®^ (cell-based tissue model) [131,132,133]	A431 cells derived from a vulval epidermoid carcinoma grown on a polycarbonate filter

## Data Availability

Not applicable.

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
