# Peer review of "Recent Advances in Studying In Vitro Drug Permeation Across Mucosal Membranes"

_pharmaceutics, 2025, doi:10.3390/pharmaceutics17020256_

Round 1
Reviewer 1 Report
Comments and Suggestions for Authors
Dear Authors,
The review article titled „Recent Advances on Studying In Vitro Drug Permeation across Mucosal Membranes” is a comprehensive overview of numerous aspects related to the mechanisms of drug permeation through mucous membranes and the methods used to evaluate the degree of drug permeation through various mucous membranes. The review focuses on four mucous membranes, namely ocular, nasal, vaginal and rectal. Sections 2 and 3, highlight the main characteristics of the aforementioned mucous membranes and their implications for drug permeation, as well as the mechanisms by which permeation can be achieved. Section 4 focuses on In vitro permeation tests (IVPT), using either tissue-based models or cell-based models, highlighting the main IVPT systems currently used and how data can be analyzed. In Section 5, future perspectives are presented, such as using non-destructive optical methods for studying drug permeation through mucous membranes. Finally, the authors briefly discuss the potential use of organ-on-chip systems and challenges that remain relevant in the field of studying drug permeation through mucous membranes and the need to establish more standardized methods in order to facilitate the development and approval of transmucosal drug products.
Observations:
1. The authors should consider expanding Section 3 regarding Drug Permeation Mechanisms across Mucosa, as well as Section 4.2 regarding Cell-based models
2. Please discuss about the artificial membranes used for IVRT and IVPT studies
3. The authors could consider moving the first paragraph regarding organ-on-chip systems from Section 6 Summary and Outlook to Section 5 Future Perspectives
Overall, the present review article adequately presented important information related to drug permeation through various mucous membranes, using illustrations and relevant examples for each mucosa.
Author Response
Comments 1: The authors should consider expanding Section 3 regarding Drug Permeation Mechanisms across Mucosa, as well as Section 4.2 regarding Cell-based models
Response 1: We thank the reviewer for the suggestions. We have expanded the Sections 3 and 4.2 as suggested (highlighted in yellow).
Comments 2: Please discuss about the artificial membranes used for IVRT and IVPT studies
Response 2: Although artificial membranes designed to mimic the blood brain barrier (BBB) and gastrointestinal tract (GIT) have been utilized in parallel aritifical membrane permeability assay (PAMPA) for IVPT studies, there are currently no artificial membranes available to mimic the mucosal membranes discussed in this paper. Since the focus of this review article is on IVPT studies, we prefer not to include a summary of artificial membranes used for IVRT studies.
Comments 3: The authors could consider moving the first paragraph regarding organ-on-chip systems from Section 6 Summary and Outlook to Section 5 Future Perspectives
Response 3: Thank you for the comment. We have restructured these two sections as suggested.

Reviewer 2 Report
Comments and Suggestions for Authors
Please include the buccal mucosa in your review.
Ethical approve concern for the use tissue and cell-based assay should be addressed and stated with update information and process.
The larger Fig. 1 with sharp and larger letter of label inside this Figure are required.
Please include the type of appropriate receptor media and pH for permeation test of four types of mucosae.
Please reveal the review on which type of cells use for permeation test and how about the comparison with the reliability to the other techniques? please provide the previously published data for this point of view covering four types of mucosa.
Please include CLSM technique into your review with its application on permeation with the prediction and evaluation the obtained data.
How about the flux value and lag time related to permeation mentioned in 4.3. Data Analysis
Please include the silico method or technique in 5. Future Perspectives
Abbreviations note should be included initially.
The reference list should be corrected and complied with the format of the journal.
The review based on patent should be included.
Schematic diagram is needed for summarize the content of "5. Future Perspectives" for more easier understanding to the reader.
Statistical analysis used for "in vitro-in vivo correlation (IVIVC)" should be addressed.
The inclusion of molecular drugs name for their name and applications should be conducted for each topic of this review.
Author Response
Comments 1: Please include the buccal mucosa in your review.
Response 1: While buccal mucosa delivery is indeed an important aspect of transmucosal drug delivery, our review specifically focuses on non-oral mucosal delivery, as outlined in the introduction. Given the scope of our paper, including buccal mucosa would extend beyond our intended focus. We appreciate your understanding.
Comments 2: Ethical approve concern for the use tissue and cell-based assay should be addressed and stated with update information and process.
Response 2: We appreciate your comment regarding ethical approval for tissue and cell-based assays and understand this is an important consideration. However, institutional and federal regulations and guidelines on using cell lines and tissues may vary. Therefore, we respectfully request not to include the information to avoid confusions.
Comments 3: The larger Fig. 1 with sharp and larger letter of label inside this Figure are required.
Response 3: We have adjusted the font size and uploaded higher resolution figure.
Comments 4: Please include the type of appropriate receptor media and pH for permeation test of four types of mucosae.
Response 4: Thank you for the comment. We have summarized the information in Table 2 as suggested.
Comments 5: Please reveal the review on which type of cells use for permeation test and how about the comparison with the reliability to the other techniques? please provide the previously published data for this point of view covering four types of mucosa.
Response 5: We have updated Table 4 with more detailed information on cell types used, and section 4.2 has been expanded to discuss different mucosal cell models reported in the literature (highlighted in yellow).
Comments 6: Please include CLSM technique into your review with its application on permeation with the prediction and evaluation the obtained data.
Response 6: The CLSM is more widely used in permeation studies across the skin, oral mucosa and intestinal mucosa. We have included the CLSM technique in Section 5 Future Perspectives (highlighted in yellow).
Comments 7: How about the flux value and lag time related to permeation mentioned in 4.3. Data Analysis
Response 7: Thank you for the comment. While lag time should be considered for flux calculation especially when lag time is long, our review primarily focuses on fundamental data analysis equations rather than detailed flux value discussion.
Comments 8: Please include the silico method or technique in 5. Future Perspectives
Response 8: Thank you for your suggestion. We have included a brief discussion on in silico methods in Section 5 due to limited information available (highlighted in yellow)
Comments 9: Abbreviations note should be included initially.
Response 9: The abbreviations section is provided at the end of the manuscript based on the journal's manuscript template.
Comments 10: The reference list should be corrected and complied with the format of the journal.
Response 10: The references were cited using the journal format in the original submission.
Comments 11: The review based on patent should be included.
Response 11: We thank the reviewer for the comment. There is limited information available in patents regarding drug permeation mechanisms and in vitro permeation testing methodologies.
Comments 12 Schematic diagram is needed for summarize the content of "5. Future Perspectives" for more easier understanding to the reader.
Response 12: Thank you for the suggestion. Since there are multiple complex future directions discussed in Section 5, we respectfully prefer not to include a schematic diagram to avoid overly simplifying these directions.
Comments 13: Statistical analysis used for "in vitro-in vivo correlation (IVIVC)" should be addressed.
Response 13: We included Jmax and total permeation amount analyses in the IVIVC section.
Comments 14: The inclusion of molecular drugs name for their name and applications should be conducted for each topic of this review.
Response 14: While we acknowledge the importance of including specific drug examples, our review concentrates on general principles and methods rather than individual drug molecules and formulations. Given the scope, we prefer not to compile an extensive database of all drugs tested, as it would shift the emphasis away from the broader scientific discussion. However, we ensure that relevant examples are incorporated where necessary to illustrate key concepts.
Round 2
Reviewer 1 Report
Comments and Suggestions for Authors
The Authors addressed all issues raised.
Author Response
Reviewer 1
The Authors addressed all issues raised.
Response: Thank you for the positive feedback.
Reviewer 2 Report
Comments and Suggestions for Authors
Please review and included on basic or principle of ethical approve concern for the use mucosal tissue and cell-based assay in 5. Future Perspectives.
The inclusion of review CLSM on permeation study is required in Topic 3 and 4.
Please describe with appropriate review on the flux value via study permeation through different mucosa since it directly relates to permeation in 4.3. Data Analysis.
The abbreviations section could be moved to the initial for easier understand to the reader.
The reference list should be corrected and complied with the format of the journal as
- Journal Articles:
1. Author 1, A.B.; Author 2, C.D. Title of the article. Abbreviated Journal Name Year, Volume, page range.
and the full all authors are required not just et al.
Please include the duration period appropriate used for different tissue and cell-based permeation experiment in 4. IVPT Considerations and Systems.
Author Response
Reviewer 2
Comment 1: Please review and included on basic or principle of ethical approve concern for the use mucosal tissue and cell-based assay in 5. Future Perspectives.
Response 1: Thank you for the comment. We have included a paragraph to summarize the basic of conducting responsible and ethical research in Section 4 (highlighted in green, before 4.3. Data Analysis).
Comment 2: The inclusion of review CLSM on permeation study is required in Topic 3 and 4.
Response 2: We respectfully disagree with the reviewer’s suggestion. Topic 3 focuses on drug permeation mechanisms across mucosal tissues, where CLSM is not relevant as a research technique. Therefore, we believe it is more appropriate to discuss CLSM in Section 5: Future Perspectives, since this technique is less established compared to the well-studied IVPT methods and apparatus summarized in Section 4. Moreover, CLSM primarily provides qualitative or semi-quantitative information, and is typically used in combination with other techniques to complement permeation studies.
Comment 3: Please describe with appropriate review on the flux value via study permeation through different mucosa since it directly relates to permeation in 4.3. Data Analysis.
Response 3: Thank you for your suggestion. We have carefully considered the reviewer's comment. In Section 4.3: Data Analysis, we focus on general principles for interpreting permeation data, including flux calculation and relevant parameters. Our aim is to provide a conceptual overview rather than an exhaustive compilation of flux data from the literature on different mucosa as many factors include drug and formulations can affect flux values.
Comment 4: The abbreviations section could be moved to the initial for easier understand to the reader.
Response 4: We agree with your comment. As we stated in the previous response to this comment, according to the journal’s manuscript template and guide for authors, the abbreviations section is listed at the end of the manuscript. If the journal’s editor approves this change, we are more than willing to move the abbreviations list to the beginning of the manuscript.
Comment 5: The reference list should be corrected and complied with the format of the journal as
Journal Articles:
- Author 1, A.B.; Author 2, C.D. Title of the article. Abbreviated Journal Name Year, Volume, page range.
and the full all authors are required not just et al.
Response 5: The reference format has been updated into “MPDI Chicago” style.
Comment 6: Please include the duration period appropriate used for different tissue and cell-based permeation experiment in 4. IVPT Considerations and Systems.
Response 6: The IVPT study duration differ across mucosal tissues and cell models depending upon testing conditions and study purposes. We have included reported IVPT testing duration for tissue-based studies in Section 4.1 (highlighted in green).